# On the Local Hessian in Back-propagation

**Huishuai Zhang**
Microsoft Research Asia
Beijing, 100080
huzhang@microsoft.com

**Wei Chen**
Microsoft Research Asia
Beijing, 100080
wche@microsoft.com

**Tie-Yan Liu**
Microsoft Research Asia
Beijing, 100080
tyliu@microsoft.com

## Abstract

Back-propagation (BP) is the foundation for successfully training deep neural networks. However, BP sometimes has difficulties in propagating a learning signal deep enough effectively, e.g., the vanishing gradient phenomenon. Meanwhile, BP often works well when combining with "designing tricks" like orthogonal initialization, batch normalization and skip connection. There is no clear understanding on what is essential to the efficiency of BP. In this paper, we take one step towards clarifying this problem. We view BP as a solution of *back-matching propagation* which minimizes a sequence of back-matching losses each corresponding to one block of the network. We study the Hessian of the local back-matching loss (*local Hessian*) and connect it to the efficiency of BP. It turns out that those designing tricks facilitate BP by improving the spectrum of local Hessian. In addition, we can utilize the local Hessian to balance the training pace of each block and design new training algorithms. Based on a scalar approximation of local Hessian, we propose a scale-amended SGD algorithm. We apply it to train neural networks with batch normalization, and achieve favorable results over vanilla SGD. This corroborates the importance of local Hessian from another side.

## 1 Introduction

Deep neural networks have been advancing the state-of-the-art performance over a number of tasks in artificial intelligence, from speech recognition [Hinton et al., 2012], computer vision [He et al., 2016a] to natural language understanding [Hochreiter and Schmidhuber, 1997]. These problems are typically formulated as minimizing non-convex objectives parameterized by the neural network models. Typically, the models are trained with stochastic gradient descent (SGD) or its variants and the gradient information is computed through back-propagation (BP) [Rumelhart et al., 1986].

It is known that BP sometimes has difficulties in propagating a learning signal deep enough effectively, e.g., the *vanishing/exploding gradient phenomenon*, [Hochreiter, 1991, Hochreiter et al., 2001]. Recent designing tricks, such as orthogonal initialization [Saxe et al., 2014], batch normalization [Ioffe and Szegedy, 2015] and skip connection [He et al., 2016a], improve the performance of deep neural networks on almost all tasks, which are interpreted to be able to alleviate the vanishing gradient to some extent. However, a recent work [Orhan and Pitkow, 2018] shows that a network with non-orthogonal skip connection always underperforms a network with orthogonal (identity is a special case) skip connection and neither network has vanishing gradient as back-propagating through layers. This suggests that vanishing gradient is not the core reason for a network being good or not. We ask that *if vanishing gradient is a superficial reason, what is essential to the efficiency of BP?*

In this paper, we consider this question from the optimization's perspective and give an answer: the Hessian of the local back-matching loss is responsible for the difficulty of training deep nets with BP. Specifically, we start from a penalized loss formulation, which takes the intermediate feature outputs as variables of the optimization and the penalty is to enforce the coordination (architecture) connection [Carreira-Perpinan and Wang, 2014]. Minimizing the penalized loss following backward

order leads to a *back-matching propagation* procedure which involves minimizing a sequence of back-matching losses. Each back-matching loss penalizes the mismatch between a target signal from the upper block and the output of the current block, which is determined by the parameters and the inputs of the block[1]. We show that BP is equivalent to minimizing each back-matching loss with one-step gradient update. This is to say, BP is a solution of the back-matching propagation procedure.

However, in general, the one-step gradient update may/may not be a good solution to minimize the back-matching loss contingent on the Hessian, as is well known that bad-conditioned Hessian can have enormous adversarial impact on the convergence of the first-order methods [Ben-Tal and Nemirovski, 2001]. Loosely speaking, if the local Hessian is badly-conditioned, the one-step gradient update does not minimize the back-matching loss sufficiently and the target signal distorts gradually when backward through layers.

We mathematically derive the formula of local Hessian and show that the designing tricks including batch normalization and skip connection can drive local Hessian towards a good-conditioned matrix to some extent. This explains why practical designing tricks can stabilize the backward process. In particular, by analyzing the local Hessian of residual block, we can answer the questions about skip connection in [Orhan and Pitkow, 2018] via local Hessian.

Besides interpreting existing practical techniques and providing guidance to design neural network structure, we can also utilize local Hessian to design new algorithms. The general idea is to employ the information of the local Hessian to facilitate the training of neural networks. We propose a scale-amended SGD algorithm to balance the training pace of each block by considering the scaling effect of local Hessian. More specifically, we approximate the local Hessian with a scalar and use the scalar to amend the gradient of each block. Such a scale-amended SGD is built upon the regular BP process, and hence it is easy to implement in current deep learning frameworks [Bastien et al., 2012, Abadi et al., 2016, Paszke et al., 2017, Seide and Agarwal, 2016]. We apply this scale-amended SGD to feed-forward networks with batch normalization and empirically demonstrate that it improves the performance by a considerable margin. This further advocates the key role of the local Hessian in efficient learning of deep neural networks.

## 1.1 Related Works

The penalty loss formulation is inspired by *methods of auxiliary coordinates (MAC)* [Carreira-Perpinan and Wang, 2014] and *proximal backpropagation* [Frerix et al., 2018]. Specifically, Carreira-Perpinan and Wang [2014] applies block coordinate descent to optimize the penalized objective. Frerix et al. [2018] applies proximal gradient when updating $W$. In contrast, we start from the penalty loss formulation and focus on the local Hessian for each subproblem in the back-matching propagation, and argue that the local Hessian is critical to the efficiency of BP.

Our scale-amended SGD is related to the algorithms tackling the difficulty of BP for training deep nets by incorporating second-order/metric information [Martens, 2010, Amari, 1998, Pascanu and Bengio, 2014, Ollivier, 2015, Martens and Grosse, 2015] and the block-diagonal second order algorithms [Lafond et al., 2017, Zhang et al., 2017, Grosse and Martens, 2016]. These second-order algorithms approximate the Hessian/Fisher matrix and are computationally expensive. In contrast, the scale-amended SGD only amends the vanilla SGD with a scalar for each block based on the approximation of local Hessian. The scale-amended SGD is closely related to the layer-wise adaptive learning rate strategy [Singh et al., 2015, You et al., 2017]. However, these two layer-wise learning rate strategies do not have explanation of why the rate is set in that way.

## 2 BP as a solution of back-matching propagation

In this section, we first introduce the quadratic penalty formulation of the loss of neural network and the *back-matching propagation* procedure which minimizes a sequence of local back-matching losses following the backward order. Then we connect BP to the one-step gradient update solution of back-matching propagation procedure. The quality of such a solution on minimizing back-matching loss is determined by its local Hessian.

**Procedure 1** Back-matching Propagation

---

**Input:** $\boldsymbol{W}_b^k$, $\boldsymbol{z}_b^k$ for $b = 1, ..., B$ and $\boldsymbol{z}_0^k = X^k$.

**for** $b = B, ..., 1$ **do**

$$\boldsymbol{W}_b^{k+1} \leftarrow \underset{\boldsymbol{W}_b}{\arg\min} \, \ell_b \left( \boldsymbol{W}_b, \boldsymbol{z}_{b-1}^k \right), \tag{3}$$

$$\boldsymbol{z}_{b-1}^{k+\frac{1}{2}} \leftarrow \underset{\boldsymbol{z}_{b-1}}{\arg\min} \, \ell_b \left( \boldsymbol{W}_b^k, \boldsymbol{z}_{b-1} \right), \tag{4}$$

**end for**
**Output:** A new parameter $\boldsymbol{W}^{k+1}$

---

Suppose the loss of training a neural network is given by[2]

$$J(\boldsymbol{W}; X, y) = \ell(y; F(\boldsymbol{W}, X)), \tag{1}$$

where $\ell(\cdot)$ is the loss function with respect to the training targets $y$ and the network output, $F(\cdot, \cdot)$ is the network mapping, $\boldsymbol{W}$ is the trainable parameter of the network and $X$ is the input data. Carreira-Perpinan and Wang [2014] introduces the intermediate output of the network as auxiliary variables and the architecture connection as a quadratic penalty, and proposes to minimize the following quadratic penalty formulation of the loss,

$$Q(\boldsymbol{W}, \boldsymbol{z}; \gamma) = \ell(\boldsymbol{y}, F_B(\boldsymbol{W}_B, \boldsymbol{z}_{B-1})) + \sum_{b=1}^{B-1} \frac{\gamma}{2} \|\boldsymbol{z}_b - F_b(\boldsymbol{W}_b; \boldsymbol{z}_{b-1})\|^2, \tag{2}$$

where $F_b(\cdot, \cdot)$ is a block mapping, $\boldsymbol{W}_b, \boldsymbol{z}_{b-1}$ are the trainable parameter and the input of network block $b$, respectively, for $b = B, ..., 1$ and $\boldsymbol{z}_0$ is the input data $X$.

It has been argued in [Nocedal and Wright, 2006] that under mild condition, the solutions of minimizing (2) converge to the solution of the original problem (1) as $\gamma \rightarrow \infty$. Carreira-Perpinan and Wang [2014] minimizes objective (2) via $z$-step and $W$-step, which is essentially a block coordinate descent algorithm.

Inspired by the form (2), we study the back-matching propagation procedure (Procedure 1), which minimizes a sequence of local back-matching losses following the backward order. The local back-matching loss for block $b$ at step $k$ is denoted by $\ell_b$

$$\ell_b(\boldsymbol{W}_b, \boldsymbol{z}_{b-1}) = \begin{cases} \ell\left(\boldsymbol{y}^k; F_B(\boldsymbol{W}_B, \boldsymbol{z}_{B-1})\right), & \text{for } b = B \\ \frac{1}{2} \left\| \boldsymbol{z}_b^{k+\frac{1}{2}} - F_b(\boldsymbol{W}_b, \boldsymbol{z}_{b-1}) \right\|^2, & \text{for } b = B-1, ..., 1, \end{cases} \tag{5}$$

where $\boldsymbol{z}_b^{k+\frac{1}{2}}$ is computed by (4) repeatedly. We note that $z^k$ is computed by forward pass given a new $X^k$ and is not updated by Procedure 1 and $z_b^{k+\frac{1}{2}}$ is an intermediate variable to store the desired change on the output $z_b$ which is used to compute $W_{b-1}^{k+1}$. For each subproblem at $b$, we alternatively optimize over $\boldsymbol{W}_b$ and $\boldsymbol{z}_{b-1}$ while fixing the other as in the forward process because jointly optimizing over $\boldsymbol{W}_b$ and $\boldsymbol{z}_{b-1}$ is non-convex even if $F_b$ represents matrix-vector product.

A direct explanation of the back-matching loss is that given the target signal $\boldsymbol{z}_b^{k+\frac{1}{2}}$ propagated from upper block, which is believed to be the direction of $\boldsymbol{z}_b^k$ to decrease the loss, the new weight $\boldsymbol{W}^{k+1}$ and the new target signal for lower block $\boldsymbol{z}_{b-1}^{k+\frac{1}{2}}$ should minimize the matching loss $\ell_b$.

We are not suggesting Procedure 1 as a new algorithm to train neural network. Actually, Procedure 1 may not be stable in practice if solving each subproblem fully [Lee et al., 2015, Wiseman et al., 2017] because the solution of (3) and (4) may deviate from last updated value too much and jump out of the trust region. Instead, we connect BP to the one-step gradient update solution of (3) and (4) and argue that the conditions of the subproblems (3) and (4) affect the efficiency of BP given the explanation of back matching loss.

**Proposition 1.** *If (3) is solved by one-step gradient update with step size $\mu$ and (4) is solved by one-step gradient update with step size 1, then $\boldsymbol{W}^{k+1}$ produced by the procedure 1 is the same as gradient update of the original problem (1) with step size $\mu$.*

*Proof.* The proof is relegated to Supplemental A due to space limit. □

We note that the form of the back-matching loss is mentioned in *target propagation* [Lee et al., 2015, Le Cun, 1986] which is motivated by the biological implausibility of BP while we formulate it from minimizing a penalized objective. We also note that the connection between BP and the one-step gradient update of minimizing (2) in backward order is made in [Frerix et al., 2018] for the case $F_b(\cdot)$ is either activation function or linear transformation.

Here we view BP as a solution of back-matching propagation and study the local Hessian matrices of back-matching losses (3) and (4),

$$\text{Local Hessian: } \boldsymbol{H}_{\text{vec}(W)} = \frac{\partial^2 \ell_b\left(\boldsymbol{W}_b, \boldsymbol{z}_{b-1}^k\right)}{\partial \text{vec}(\boldsymbol{W})^2}, \quad \boldsymbol{H}_z = \frac{\partial^2 \ell_b\left(\boldsymbol{W}_b^k, \boldsymbol{z}_{b-1}\right)}{\partial \boldsymbol{z}_{b-1}^2}. \tag{6}$$

The Hessian of training deep neural networks has been studied in previous works Dauphin et al. [2014], Orhan and Pitkow [2018], Li et al. [2016], Sagun et al. [2017], Jastrzębski et al. [2018]. They all analyze and calculate the Hessian of the objective with respect to the whole network parameter. In contrast, we study the Hessian of the local back-matching loss and connect it to the efficiency of BP.

Loosely speaking, if the local Hessian of (5) with respect to $\boldsymbol{W}$ is good-conditioned, the solution of (3) minimizes the local back-matching loss sufficiently, which implies that the target signal is efficiently approximated by updating parameters of current block, and if the local Hessian of (5) with respect to $\boldsymbol{z}$ is good-conditioned, the solution of (4) minimizes the local back-matching loss sufficiently, which implies that the target signal is efficiently back-propagated. Next, we show how skip connection and batch normalization improve the spectrum of the local Hessian.

## 3 Explain the efficiency of BP via local Hessian

Because the condition of local Hessian determines how efficiently the back-matching loss is minimized by updating the parameters of current block and how accurately the error signal propagates back to the lower layer, we evaluate how good a block is via analyzing its local Hessian. We first analyze the local Hessian of a fully connected layer[3] and then show that the skip connection and batch normalization improve the spectrum of local Hessian and hence facilitate the efficiency of BP.

### 3.1 Block of a fully connected layer

We consider a block $b$ composed of a fully connected layer with $n_b$ outputs and $n_{b-1}$ inputs. The mapping function is given by

$$\boldsymbol{z}_b = F_b(\boldsymbol{W}_b, \boldsymbol{z}_{b-1}) = \boldsymbol{W}_b \cdot \boldsymbol{z}_{b-1}, \tag{7}$$

where $\boldsymbol{W}_b$ is an $n_b \times n_{b-1}$ matrix.

Suppose that after the gradient step on $\boldsymbol{z}_b$ from upper layer, we get an intermediate variable $\boldsymbol{z}_b^{k+1/2}$. The Hessian of back matching loss (5) with respect to $\boldsymbol{z}_{b-1}$ and $\boldsymbol{w}_b$ are

$$\boldsymbol{H}_z = (\boldsymbol{W}_b^k)^T \boldsymbol{W}_b^k, \tag{8}$$

$$\boldsymbol{H}_w = \sum_{j=1}^m \boldsymbol{z}_{b-1}^k[j](\boldsymbol{z}_{b-1}^k[j])^T, \tag{9}$$

respectively, where $m$ is the batch size, $[j]$ represents the $j$-th sample and $\boldsymbol{w}_b$ is a vector of a row of $\boldsymbol{W}_b$. Then $\boldsymbol{H}_{\text{vec}(W_b)}$ is a block diagonal matrix with each block being $\boldsymbol{H}_w$ where $\text{vec}(\boldsymbol{W}_b)$ is a long vector stacking the rows of $\boldsymbol{W}_b$ first.

For (8) the local Hessian with respect to $\boldsymbol{z}_{b-1}$, we derive the distribution of its eigenvalues. For the convenience of analysis, we assume the elements of $\boldsymbol{w}_b$ are independently generated from Gaussian distribution with mean 0 and variance $\sigma^2$ and $n_b, n_{b-1} \to \infty$ and the ratio $n_b/n_{b-1} \to c \in (0, +\infty)$.

Then by the Marchenko-Pastur law [Marčenko and Pastur, 1967], we have the density of the eigenvalue $\lambda$ of (8) as follows,

$$\nu(A) = \begin{cases} (1-c)\mathbf{1}_{0 \in A} + \nu_2(A), & \text{if } 0 < c \le 1, \\ \nu_2(A), & \text{if } c > 1, \end{cases} \tag{10}$$

where

$$d\nu_2(\lambda) = \frac{c}{2\pi\sigma^2} \frac{\sqrt{(c_+ - \lambda)(\lambda - c_-)}}{\lambda} \mathbf{1}_{[c_-, c_+]} d\lambda, \tag{11}$$

with $c_+ = \sigma^2(1 + \sqrt{c})^2/c, c_- = \sigma^2(1 - \sqrt{c})^2/c$.

This result affirms that the orthonormal initialization [Mishkin and Matas, 2016, Saxe et al., 2014] facilitates backward propagation. If $\boldsymbol{W}_b$ is an orthonormal matrix, the eigenvalues of $\boldsymbol{H}_z$ are composed of $n_b$ 1's and $n_{b-1} - n_b$ 0's if $n_b < n_{b-1}$. This is the best spectrum of Hessian we can expect for minimizing the back-matching loss (5).

However, in general, $\boldsymbol{W}_b$ is not orthonormal and hence $\boldsymbol{H}_z$ is not identity. The gradient update on $\boldsymbol{z}_{b-1}$ does not minimize the back-matching loss well. As back propagating to lower blocks, the update $\boldsymbol{z}_{b-t}^{k+\frac{1}{2}} - \boldsymbol{z}_{b-t}^k$ gets far from the direction of minimizing the back-matching loss $\ell_b$ for $t = 1, 2, ..., b$. Such discrepancy becomes larger as the condition of $\boldsymbol{H}_z$ of each block is bad and as the back-propagation goes deep.

For the local Hessian with respect to $\boldsymbol{W}$, it is hard to control in general. Several recent works [Frerix et al., 2018, Ye et al., 2017] suggest using forms involving $\boldsymbol{H}_W$ to precondition vanilla SGD. We note that Le Cun et al. [1991] has also studied the spectrum of $\boldsymbol{H}_w$ which gives a theoretical justification for the choice of centered input over biased state variables.

We next study the local Hessian of blocks with skip connection and batch normalization and show that these designing tricks can improve the spectrum of $\boldsymbol{H}_z$ and $\boldsymbol{H}_W$ to some extent and hence make the training deep neural networks easier.

## 3.2 Block with skip connection

Skip connection has been empirically demonstrated important to obtain state-of-the-art results [He et al., 2016a,b, Huang et al., 2017a], while its functionality has various interpretations. Veit et al. [2016] argue that residual network can be seen as an ensemble of shallow nets and avoids vanishing gradient problem by introducing short paths. Jastrzebski et al. [2018] suggest that residual block performs iterative refinement of features for higher layer while lower layers concentrate representation learning behavior. These works focus on the interpretation of how Resnet works. We here try to give an answer on why Resnet works from the optimization perspective. A recent work [Orhan and Pitkow, 2018] argues that skip connection eliminates singular points of the Hessian matrix and there are open questions in [Orhan and Pitkow, 2018], for which we can give answers by analyzing the local Hessian of residual block.

Suppose that the mapping of residual block is given by

$$\boldsymbol{z}_b = F_b(\boldsymbol{W}_b, \boldsymbol{z}_{b-1}) = \boldsymbol{z}_{b-1} + \phi_b(\boldsymbol{W}_b, \boldsymbol{z}_{b-1}), \tag{12}$$

where $F_b(\cdot)$ is the residual block mapping with parameters $\boldsymbol{W}_b$ and input $\boldsymbol{z}_{b-1}$. The Hessians of the back-matching loss (5) with respect to $\boldsymbol{z}_{b-1}$ and $\boldsymbol{W}_b$ are given by

$$\boldsymbol{H}_z = \left(\boldsymbol{I} + \frac{\partial\phi_b}{\partial\boldsymbol{z}_{b-1}}\right)^T \left(\boldsymbol{I} + \frac{\partial\phi_b}{\partial\boldsymbol{z}_{b-1}}\right) - \frac{\partial}{\partial\boldsymbol{z}_{b-1}}\left(\frac{\partial F_b}{\partial\boldsymbol{z}_{b-1}} \cdot \left(\boldsymbol{z}_b^{k+\frac{1}{2}} - \boldsymbol{z}_b^k\right)\right), \tag{13}$$

$$\boldsymbol{H}_W = \left(\frac{\partial F_b}{\partial\text{vec}(\boldsymbol{W}_b)}\right)^T \left(\frac{\partial F_b}{\partial\text{vec}(\boldsymbol{W}_b)}\right) - \frac{\partial}{\partial\text{vec}(\boldsymbol{W}_b)}\left(\frac{\partial F_b}{\partial\text{vec}(\boldsymbol{W}_b)} \cdot \left(\boldsymbol{z}_b^{k+\frac{1}{2}} - \boldsymbol{z}_b^k\right)\right) \tag{14}$$

We can see that (14) the Hessian of local matching loss for residual block with respect to $\boldsymbol{W}_b$ is the same as the case without skip connection. Thus we focus on (13) the local Hessian with respect to $\boldsymbol{z}$. Specifically, we analyze the first part of (13), the Gauss-Newton matrix, which is a good positive semidefinite approximation to the Hessian [Martens, 2016, Chen, 2011]. Define the condition number of a matrix $\boldsymbol{M}$ as $C(\boldsymbol{M}) := \sigma_{\max}(\boldsymbol{M})/\sigma_{\min}(\boldsymbol{M})$, where $\sigma_{\max}$ and $\sigma_{\min}$ are the largest and smallest non-zero singular values, respectively. The larger the condition number, the worse the problem.

**Remark 1.** *If a)* $\frac{\partial \phi_b}{\partial z_{b-1}}$ *is "small" relatively i.e.,* $\sigma_{\max}\left(\frac{\partial \phi_b}{\partial z_{b-1}}\right) < 1 - s$ *for some constant* $s > 0$, *and b)* $C\left(\frac{\partial \phi_b}{\partial z_{b-1}}\right) > \frac{1+s}{1-s}$, *then*

$$C\left(\boldsymbol{I} + \frac{\partial \phi_b}{\partial z_{b-1}}\right) < C\left(\frac{\partial \phi_b}{\partial z_{b-1}}\right). \tag{15}$$

This indicates that the condition number of the Gauss-Newton matrix with skip connection is guaranteed to be smaller than that without skip connection given two assumptions. The assumption **b)** is generally satisfied for neural network from the spectrum distribution analysis of fully-connected layer in Section 3.1 while the assumption **a)** seems a bit strong. We cannot verify assumption **a)** analytically because $\phi_b(\cdot)$ typically involves more than two linear layers, nonlinear activations and batch normalization layers. We leave the empirical study on the spectrum distribution of local Hessian of the residual block for future work.

Interestingly, Orhan and Pitkow [2018] demonstrate that a network with an orthogonal connection achieves the performance as good as the one with identity skip connection, which can be easily explained from the fact that orthogonal skip connection does not change the condition number of the local Gauss-Newton matrix (the first part of (13)). Furthermore, Orhan and Pitkow [2018] also empirically show that a network with non-orthogonal skip connection always underperforms a network with orthogonal (identity is a special case) skip connection though neither network has vanishing gradient as back-propagating through layers. This can be easily argued from the formula (13) as non-orthogonal skip connection has larger condition number than orthogonal skip connection whose eigenvalues are all 1's.

## 3.3 Block with batch normalization

Batch normalization (BN) is widely used for accelerating the training of feed-forward neural networks. In this section, we consider adding a BN layer after a fully-connected layer. We fix the affine transformation of BN to be identity for simplicity. If $z_b^k$ represents one component of $\boldsymbol{z}_b^k$ and $\boldsymbol{w}_b$ a vector of one row of $\boldsymbol{W}_b$, then the BN layer mapping is given by

$$z_b^k = \text{BN}\left(\tilde{z}_b^k\right) = \left(\tilde{z}_b^k - \mathbb{E}[\tilde{z}_b^k]\right)/\sqrt{\text{Var}[\tilde{z}_b^k]}, \quad \text{where} \quad \tilde{z}_b^k = (\boldsymbol{w}_b)^T \boldsymbol{z}_{b-1}. \tag{16}$$

BP through a fully connected layer with BN is given in [Ioffe and Szegedy, 2015] and we provide the form for the back matching loss in Supplemental B for completeness. The gradient formula is quite complicated, as the $\mathbb{E}$ and $\text{Var}$ involve batch information. To proceed the analysis, we ignore the terms involving $1/m$, which does not lose much as the batch size becomes large.

Now we compute the local Hessian of the fully connected layer with BN as follows[4]

$$\boldsymbol{H}_z \approx \sum_{i=1}^{n_b} \frac{\boldsymbol{w}_b^k(i) \cdot \boldsymbol{w}_b^k(i)^T}{\text{Var}[\tilde{z}_b^k(i)]} = \sum_{i=1}^{n_b} \frac{\boldsymbol{w}_b^k(i) \cdot \boldsymbol{w}_b^k(i)^T}{\text{Var}[\boldsymbol{w}_b^k(i)^T \boldsymbol{z}_{b-1}^k]}, \tag{17}$$

$$\boldsymbol{H}_w \approx \frac{\sum_{j=1}^m \boldsymbol{z}_{b-1}^k[j](\boldsymbol{z}_{b-1}^k[j])^T}{\text{Var}[\tilde{z}_b^k(i)]} = \frac{\sum_{j=1}^m \boldsymbol{z}_{b-1}^k[j](\boldsymbol{z}_{b-1}^k[j])^T}{\text{Var}[\boldsymbol{w}_b^k(i)^T \boldsymbol{z}_{b-1}^k]}, \tag{18}$$

where $n_b$ is the number of outputs of layer $b$, $\boldsymbol{w}_b^k(i)$ is the vector of the $i$-th row of $\boldsymbol{W}_b^k$, and $\boldsymbol{z}_{b-1}^k[j]$ represents the input of the block $b$ of the sample $j$. We next show how BN facilitates BP for training deep networks.

We first derive the distribution of the eigenvalues of (17) and compare it to (10) (the case without BN). Our assumption on $\boldsymbol{w}_b$ is the same as the one to derive (11). In contrast to that $\boldsymbol{H}_z$ being the sum of outer products of Gaussian vectors in Section 3.1, here $\boldsymbol{H}_z$ is the sum of the outer products of $\boldsymbol{w}_b/\|\boldsymbol{w}_b\|$'s which are the unit vectors equally distributed on the sphere. The density of the eigenvalue $\lambda$ of (17) is of the form (10) with [Marčenko and Pastur, 1967],

$$d\nu_2(\lambda) = \frac{\sqrt{(c_+ - \lambda)(\lambda - c_-)}}{2\pi\lambda} \mathbf{1}_{[c_-, c_+]} d\lambda, \tag{19}$$

where $c_+ = (1 + \sqrt{c})^2, c_- = (1 - \sqrt{c})^2$.

**Remark 2.** *Scaling the variance of the block parameter does not affect the spectrum of $\boldsymbol{H}_z$ in* (17).

This is in contrast to (8) where the spectrum is linearly scaled with the variance of weight parameters. Thus BP gains benefit because it acts as one-step gradient update with fixed step size 1 for all blocks.

Another benefit of BN is to improve the condition of $\boldsymbol{H}_w$ if $\boldsymbol{z}_{b-1}$ is the output of a BN .

**Remark 3.** *If $\boldsymbol{z}_{b-1}$ is the output of BN and $\boldsymbol{w}_b$ is independent of $\boldsymbol{z}_{b-1}$, then $\mathbb{E}\mathrm{diag}(\boldsymbol{H}_w) = \boldsymbol{I}/\|\boldsymbol{w}_b\|^2$.*

This indicates the problem (3) is well-conditioned and hence large step size is allowed [Ioffe and Szegedy, 2015].

## 4 Utilize local Hessian: An example

As previous section has shown the importance of the local Hessian, this section discuss how to utilize local Hessian to improve the performance on current deep learning tasks. One direct way of using local Hessian is to design better architecture. The spectrum of local Hessian can be a criteria to determine whether a building block is good or not for BP. One potential usage of local Hessian could be in neural architecture search [Zoph and Le, 2016]. As most of the time in neural architecture search is used to train huge amount of small networks and it will greatly accelerate if using local Hessian to prune the search space.

Another direction is to utilize the local Hessian to design new algorithms to improve the training of existing neural networks. Several works can be understood as examples, e.g., proximal propagation [Frerix et al., 2018] and Riemannian approaches [Cho and Lee, 2017, Huang et al., 2017b].

In this section, we propose a way to employ the information of the local Hessian to facilitate the training of deep neural networks. Ideally, good alternatives to minimize back-matching loss $\ell_b$ are $\boldsymbol{H}_W^{-1}\delta\boldsymbol{w}_b$ and $\boldsymbol{H}_z^{-1}\delta\boldsymbol{z}_{b-1}$, where $\delta\boldsymbol{w}_b$ and $\delta\boldsymbol{z}_{b-1}$ are the gradient computed via BP rule given $\boldsymbol{z}^{k+\frac{1}{2}} - \boldsymbol{z}^k$. However, $\boldsymbol{H}_W$ and $\boldsymbol{H}_z$ are often indefinite and expensive to compute. We suggest using two scalars $m_{b,W}$ and $m_{b,z}$ to evaluate how $\boldsymbol{H}_W$ and $\boldsymbol{H}_z$ scale the norm of a vector with general position, respectively. Then the back-matching loss can be approximated as

$$\ell_b\left(\boldsymbol{W}_b, \boldsymbol{z}_{b-1}^k\right) \approx \ell_b\left(\boldsymbol{W}_b^k, \boldsymbol{z}_{b-1}^k\right) + \left\langle \frac{\partial\ell_b}{\partial\boldsymbol{W}_b^k}, \boldsymbol{W}_b - \boldsymbol{W}_b^k \right\rangle + \frac{1}{2}(\boldsymbol{W}_b - \boldsymbol{W}_b^k)^T\boldsymbol{H}_W(\boldsymbol{W}_b - \boldsymbol{W}_b^k)$$

$$\approx \ell_b\left(\boldsymbol{W}_b^k, \boldsymbol{z}_{b-1}^k\right) + \left\langle \frac{\partial\ell_b}{\partial\boldsymbol{W}_b^k}, \boldsymbol{W}_b - \boldsymbol{W}_b^k \right\rangle + \frac{1}{2}m_{b,W}\|\boldsymbol{W}_b - \boldsymbol{W}_b^k\|_2^2, \tag{20}$$

$$\ell_b\left(\boldsymbol{W}_b^k, \boldsymbol{z}_{b-1}\right) \approx \ell_b\left(\boldsymbol{W}_b^k, \boldsymbol{z}_{b-1}^k\right) + \left\langle \frac{\partial\ell_b}{\partial\boldsymbol{z}_{b-1}^k}, \boldsymbol{z}_{b-1} - \boldsymbol{z}_{b-1}^k \right\rangle + \frac{1}{2}m_{b,z}\|\boldsymbol{z}_{b-1} - \boldsymbol{z}_{b-1}^k\|_2^2, \tag{21}$$

where the approximation is composed of a second-order Taylor expansion and a scaling effect of local Hessian, and $\boldsymbol{W}_b$ may represent $\mathrm{vec}(\boldsymbol{W}_b)$ contingent on the context.

We next propose an algorithm *scale-amended SGD* to take the effect of $m_{b,W}$ and $m_{b,z}$ into account to balance the training pace of each block. *Scale-amended SGD* uses $m_{b,W}$ and $m_{b,z}$ to amend the scale of vanilla BP of each block. We set the initial *backward factor* of the output layer $m = 1$, which indicates that the derivative of the loss with respect to the output of the network is regarded as the desired changes on the output to minimize the loss.

Then following the backward order, if a block has parameter $\boldsymbol{W}_b$ and gradient $\delta\boldsymbol{W}_b$ computed by BP, then we use $\delta'\boldsymbol{W}_b := \delta\boldsymbol{W}_b/m/m_{b,W}$ as the scale-amended gradient to update $\boldsymbol{W}_b$, where $m$ is the backward factor on the output of the block and $m_{b,W}$ is the scalar used to approximate $\boldsymbol{H}_{b,W}$. Then we update the backward factor $m$ for next block via $m \leftarrow m \cdot m_{b,z}$, where $m_{b,z}$ is the scalar used to approximate $\boldsymbol{H}_{b,z}$. This strategy is described in Algorithm 2.

### 4.1 Scale-amended SGD for feed-forward networks with BN

Note that for the feed-forward networks with BN layers, we can obtain a reliable estimation of $m_{b,W}$ and $m_{b,z}$. Specifically, we assume that $\boldsymbol{W}_b$ is row homogeneous [Ba et al., 2016], i.e., they represent the same level of information and are roughly of similar magnitude, and define

$$\|\boldsymbol{W}_b\|_{2,\mu}^2 := \frac{1}{\#row(\boldsymbol{W}_b)} \sum_{i=1}^{\#row(\boldsymbol{W}_b)} \boldsymbol{w}_b(i)^T\boldsymbol{w}_b(i),$$

**Algorithm 2** Scale-amended SGD
***
**Input: Gradient** $\delta \boldsymbol{W}_b$ **and scaling factor** $m_{b,W}, m_{b,z}$. **for** $b = 1, ..., B$; **Initialize** $m = 1$.

**for** $b = B, ..., 1$ **do**
$$\delta' \boldsymbol{W}_b \leftarrow \delta \boldsymbol{W}_b / m / m_{b,W} \qquad (22)$$
$$m \leftarrow m \cdot m_{b,z} \qquad (23)$$

**end for**
***

where $\boldsymbol{w}_b(i)$ is the $i$-th row of $\boldsymbol{W}_b$. Under this assumption, the scalars to approximate the local Hessians (17) and (18) of the fully connected layer with BN are computed as follows,

$$m_{b,z} := \|\boldsymbol{W}_b^T\|_{2,\mu}^2 / \|\boldsymbol{W}_b\|_{2,\mu}^2, \qquad m_{b,W} := 1/\|\boldsymbol{W}_b\|_{2,\mu}^2. \qquad (24)$$

We next evaluate the *scale-amended SGD* on training VGG nets [Simonyan and Zisserman, 2015] for image classification tasks with two datasets: CIFAR-10 [Krizhevsky and Hinton, 2009] and CIFAR-100 [Krizhevsky and Hinton, 2009]. We modify the VGG nets by keeping the last fully connected layers and removing the intermediate two fully connected layers and all the biases. Each intermediate layer of the VGG nets concatenates a BN layer right before the activation function and the BN has no trainable parameters.

During training, the images of CIFAR-10 and CIFAR-100 datasets are randomly flipped and rotated for data augmentation. The hyper-parameters for vanilla SGD and our scale-amended SGD are the same including learning rate $\eta = 0.1$ (because the backward factor for linear layer of CIFAR10 is around $\frac{10}{512}$, small learning rate $\eta = 0.005$ works better for CIFAR10 to use scale-amened SGD), momentum 0.9 and weight decay[5] coefficient 0.005. We reduce the learning rate by half once the validation accuracy is on plateau (ReduceLROnPlateau in PyTorch with patience=10), which works well for both vanilla-SGD and scale-amended SGD.

We compare the learning curves between *scale-amended SGD* and vanilla SGD on training VGG13 for CIFAR10 and CIFAR-100 classification tasks. Two algorithms start from the same initialization and pass the same batches of data. Both algorithms are run 300 epochs. We plot the learning curves in Figure 1. From Figure 1, we can see that the learning curves of our algorithm and SGD have

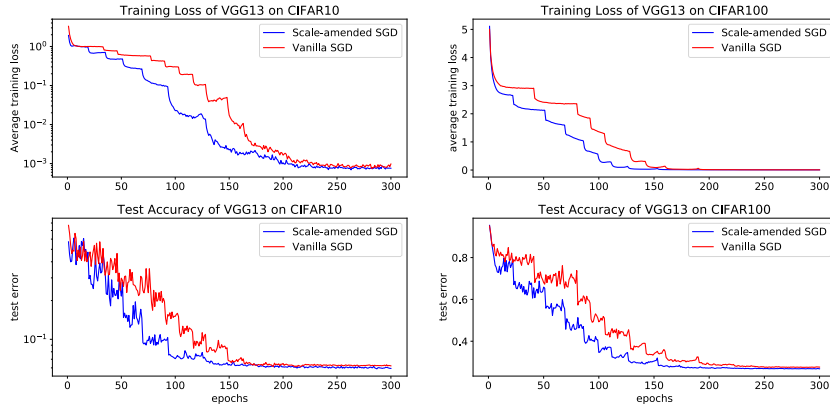

Figure 1: Comparison of vanilla SGD and *scale-amended SGD* on training VGG13 for CIFAR10 and CIFAR-100 classification. Hyperparameters are the same: learning rate 0.1 (except for CIFAR10 scale-amended SGD uses 0.005), momentum 0.9, weight decay 0.005.

similar trend (we plot curves of multiple runs and their average in Supplemental D). This is because *scale-amended SGD* only modifies the magnitude of each block gradient as a whole and does not involve any further information (second order information) and hyper-parameters are the same for both algorithms. Scrutinizing more closely, we can see our training loss curve is almost always lower than SGD's and our test error ends with a considerably lower number. Thus the scale-amended SGD

achieves favorable result over vanilla SGD on training feed-forward neural network with BN. More extensive experiments can be found in Supplemental D.

## 5 Conclusion

In this paper we view BP as a solution of *back-matching propagation* which minimizes a sequence of back-matching losses. By studying the Hessian of the local back-matching loss, we interpret the benefits of practical designing tricks, e.g., batch normalization and skip connection, in a unified way: improving the spectrum of local Hessian. Moreover, we propose scale-amended SGD algorithm by employing the information of local Hessian via a scalar approximation. Scale-amended SGD achieves favorable results over vanilla SGD empirically for training feed-forward networks with BN, which corroborates the importance of local Hessian.

**Acknowledgments**

The authors would like to thank Prof. Yuejie Chi for helpful discussion.

## Footnotes

[1]Here a block can be composed of one layer or multiple layers.

[2]For simplicity we omit the bias term in the sequel.

[3]The formula for convolution layer is given in Supplemental C.

[4]We ignore the terms involving $1/m$ again.

[5]For *scale-amended SGD*, we first apply the weight decay and then amend the scale.

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
