[Supplementary Material · supplementary.pdf]

# Supplementary Material

## A   Proof for Proposition 1

The proof is straightforward. The key is to show the one-step gradient update of

$$z_{b-1}^{k+\frac{1}{2}} \leftarrow \underset{z_{b-1}}{\arg\min} \, \ell_b \left( \boldsymbol{W}_b^k, \boldsymbol{z}_{b-1} \right) \tag{1}$$

with step size 1 satisfies

$$z_{b-1}^{k+\frac{1}{2}} - z_{b-1}^k = -\left. \frac{\partial J}{\partial \boldsymbol{z}_{b-1}} \right|_{\boldsymbol{W}=\boldsymbol{W}^k, \boldsymbol{z}=\boldsymbol{z}^k}, \tag{2}$$

where $J$ is given by Equation 1.

For the case $b = B$, we have $\ell_b(\boldsymbol{W}_b, \boldsymbol{z}_{b-1}) = \ell \left( \boldsymbol{y}^k; F_B(\boldsymbol{W}_B, \boldsymbol{z}_{B-1}) \right)$, and the one-step gradient solution of (1) is

$$z_{B-1}^{k+\frac{1}{2}} = z_{B-1}^k - 1 \cdot \left. \frac{\partial \ell_B}{\partial \boldsymbol{z}_{B-1}} \right|_{\boldsymbol{W}_B=\boldsymbol{W}_B^k, \boldsymbol{z}_{B-1}=\boldsymbol{z}_{B-1}^k}, \tag{3}$$

which satisfies (2).

For other cases of $b$, we have $\ell_b(\boldsymbol{W}_b, \boldsymbol{z}_{b-1}) = \frac{1}{2} \left\| \boldsymbol{z}_b^{k+\frac{1}{2}} - F_b(\boldsymbol{W}_b, \boldsymbol{z}_{b-1}) \right\|^2$. Suppose the one-step gradient solution of (1) satisfies (2) for some $b$. We next verify it for the case $b - 1$. Since

$$z_{b-2}^{k+\frac{1}{2}} = z_{b-2}^k - 1 \cdot \left. \frac{\partial \ell_{b-1}}{\partial \boldsymbol{z}_{b-2}} \right|_{\boldsymbol{W}_{b-1}=\boldsymbol{W}_{b-1}^k, \boldsymbol{z}_{b-2}=\boldsymbol{z}_{b-2}^k}, \tag{4}$$

then

$$
\begin{aligned}
z_{b-2}^{k+\frac{1}{2}} - z_{b-2}^k &= -\left. \frac{\partial \ell_{b-1}}{\partial \boldsymbol{z}_{b-2}} \right|_{\boldsymbol{W}_{b-1}=\boldsymbol{W}_{b-1}^k, \boldsymbol{z}_{b-2}=\boldsymbol{z}_{b-2}^k} \\
&= \left( \left. \frac{\partial F_{b-1}}{\partial \boldsymbol{z}_{b-2}} \right|_{\boldsymbol{W}_{b-1}=\boldsymbol{W}_{b-1}^k, \boldsymbol{z}_{b-2}=\boldsymbol{z}_{b-2}^k} \right)^T \cdot (z_{b-1}^{k+\frac{1}{2}} - z_{b-1}^k) \\
&= -\left. \left( \frac{\partial F_{b-1}}{\partial \boldsymbol{z}_{b-2}} \cdot \frac{\partial J}{\partial \boldsymbol{z}_{b-1}} \right) \right|_{\boldsymbol{W}=\boldsymbol{W}^k, \boldsymbol{z}=\boldsymbol{z}^k} \\
&= -\left. \frac{\partial J}{\partial \boldsymbol{z}_{b-2}} \right|_{\boldsymbol{W}=\boldsymbol{W}^k, \boldsymbol{z}=\boldsymbol{z}^k}.
\end{aligned}
$$

(5)

(6)

Following chain rule, this completes the proof.

## B   BP through BN for back-matching loss

The gradient for the back matching loss of a fully connected layer with BN is given by

$$\frac{\partial \ell_b}{\partial z_b} = z_b - z_b^{k+\frac{1}{2}}, \tag{7}$$

$$\frac{\partial \ell_b}{\partial \tilde{z}_b} = \frac{\partial \ell_b}{\partial z_b} \cdot \frac{1}{\sqrt{\mathsf{Var}[\tilde{z}_b]}} + \frac{\partial \ell_b}{\partial \mathsf{Var}[\tilde{z}_b]} \cdot \frac{2(\tilde{z}_b - \mathbb{E}[\tilde{z}_b])}{m} + \frac{1}{m} \cdot \frac{\partial \ell_b}{\partial \mathbb{E}[\tilde{z}_b]}, \tag{8}$$

$$\frac{\partial \ell_b}{\partial z_{b-1}} = (\boldsymbol{W}_b^k)^T \frac{\partial \ell_b}{\partial \tilde{z}_b}, \tag{9}$$

$$\frac{\partial \ell_b}{\partial \boldsymbol{w}_b} = \frac{\partial \ell_b}{\partial \tilde{z}_b} \cdot (\boldsymbol{z}_{b-1}^k)^T, \tag{10}$$

where $m$ is the mini-batch size, and $\frac{\partial \ell_b}{\partial \mathsf{Var}[\tilde{z}_b]}$ and $\frac{\partial \ell_b}{\partial \mathbb{E}[\tilde{z}_b]}$ is the gradient on quantities $\mathsf{Var}[\tilde{z}_b]$ and $\mathbb{E}[\tilde{z}_b]$ respectively.

## C   Local Hessian for convolutional layer

In this part we derive the Hessian of the back-matching loss for a convolutional layer. We change the notation a bit for clear representation. The weight parameter $\boldsymbol{W}$ is an array with dimension $n \times m \times w \times h$, where $n$ and $m$ are the number of output features and the number of input features respectively, and $w$ and $h$ are the width and height of convolutional kernels. Suppose the output feature size is $q_1 \times q_2$ and the input feature size is $p_1 \times p_2$. We use $b_{ku_1u_2}$ to denote the output at location $(u_1, u_2)$ of feature $k$ and $a_{ju_1u_2}$ to denote the input at location $(u_1, u_2)$ of feature $j$, then the forward process is

$$b_{ku_1u_2} = \sum_{j=1}^{n} \sum_{v_1v_2} a_{j(u_1+v_1)(u_2+v_2)} w_{jkv_1v_2}, \tag{11}$$

and the BP is given by

$$\delta a_{ju_1u_2} = \sum_{k=1}^{m} \sum_{v_1v_2} \delta b_{k(u_1+v_1)(u_2+v_2)} w_{jkv_1v_2}, \tag{12}$$

$$\delta w_{jkv_1v_2} = \sum_{u_1u_2} \delta b_{ku_1u_2} a_{j(u_1+v_1)(u_2+v_2)}. \tag{13}$$

However, this formula of the forward and backward process of convolutional layer make the derivation of Hessian complex. Note that the convolution operation essentially performs dot products between the convolution kernels and local regions of the input. The forward pass of a convolution layer can be formulated as one big matrix multiply with *im2col* operation. In order to describe back matching process clearly, we rewrite the convolution layer forward and backward pass with *im2col* operation. We use $\boldsymbol{W}_{row}$ and $\boldsymbol{W}_{col}$ to represent the weight matrices with dimension $n \times (mwh)$ and $m \times (nwh)$, respectively, which both are stretched out from $\boldsymbol{W}(n, m, w, h)$. To mimic the convolutional operation, we rearrange the input features $\boldsymbol{a}$ into a big matrix $\boldsymbol{a}_{i2c}$ through *im2col* operation: each column of $\boldsymbol{z}_{i2c}$ is composed of the elements of $\boldsymbol{a}$ that are used to compute one location in $\boldsymbol{b}$. Thus if $\boldsymbol{b}$ has dimension $n \times q_1 \times q_2$, then $\boldsymbol{a}_{i2c}$ has dimension $mwh \times q_1q_2$. Furthermore, we stack the latter two dimensions of $\boldsymbol{b}$ into a tall vector, denoted as $\boldsymbol{b}_{col}$ which has dimension $n \times q_1q_2$. The forward process (11) of convolutional layer can be rewritten as

$$\boldsymbol{b}_{col} = \boldsymbol{W}_{row} \boldsymbol{a}_{i2c} \tag{14}$$

Similarly, we can rewrite the regular BP (12) and (13) as

$$\delta a_{ju_1u_2}(x) = \boldsymbol{w}_{ju_1u_2\rightarrow}^T \delta \boldsymbol{b}(x), \tag{15}$$

$$\delta \boldsymbol{W}_{row} = \mathbb{E}_x \delta \boldsymbol{b}_{col}(x) \boldsymbol{z}_{i2c}^T(x), \tag{16}$$

where $\boldsymbol{w}_{ju_1u_2\rightarrow}$ is a vector of dimension $nq_1q_2$, whose non-zero elements are those weights that interact with input location $ju_1u_2$. There are approximately $n \times wh/c$ non-zero elements and $c$ is a factor related with pooling, padding and stride (if padding=same-size, stride=2, then c=4). The non-zero elements are scattered into $n$ blocks, with each block $wh/c$ non-zero elements, whose location within the block is corresponding to $(u_1, u_2)$. With these notations, we can derive the formula of local Hessian, given by

$$\boldsymbol{H}_{W_n} = \mathbb{E} \boldsymbol{a}_{i2c} \boldsymbol{a}_{i2c}^T, \tag{17}$$

$$\boldsymbol{H}_a(j, u_1, u_2, k, v_1, v_2) = \frac{\partial^2 \ell}{\partial a_{ju_1u_2} a_{kv_1v_2}} = \boldsymbol{w}_{ju_1u_2\rightarrow}^T \boldsymbol{w}_{kv_1v_2\rightarrow}, \tag{18}$$

$$\boldsymbol{H}_a = \boldsymbol{W}_a^T \boldsymbol{W}_a, \tag{19}$$

where $\boldsymbol{W}_a$ is a $nq_1q_2 \times mp_1p_2$ matrix each column being $\boldsymbol{w}_{ju_1u_2\rightarrow}$. We know $\boldsymbol{W}_a$ is a sparse matrix and so is $\boldsymbol{H}_a$. Moreover, $\boldsymbol{H}_{W_n}$ is a concentrated matrix as each component is a summation of $q_1q_2 \times$ batch-size variables. As the convolutional layer is essentially a linear mapping, the formulas here is similar to those of the fully connected layer although they are more involved.

### C.1   Approximate convolution layer with BN

We approximate $\boldsymbol{H}_{W_n}$ by a scalar $m_{b,W_n} = s/\|\boldsymbol{W}_{row}\|_{2,\mu}^2$, where $s = q_1q_2$ is the sharing parameter and $q_1 \times q_2$ is the output feature size.

We approximate $\boldsymbol{H}_a$ by a scalar $m_{b,z} = \|\boldsymbol{W}_{col}\|_{2,\mu}^2/\|\boldsymbol{W}_{row}\|_{2,\mu}^2/c$, where $c$ is a factor related with pooling, padding and stride (if padding=same-size, stride=2, then c=4).

Figure 1: Multiple runs of Figure 1 for CIFAR100.

## D  Other experiments

In order to verify the stability of scale-amended SGD, we run and plot multiple times of the learning curves as in Figure 1 here. We do extensive experiments to verify the effectivity of scale-amended SGD on training feed-forward neural networks with BN. First we introduce several baseline algorithms and their settings.

The first base algorithm is the vanilla SGD with *Nesterov momentum* 0.9. The learning rate is chosen to be $\eta = 0.1$ given a pool of candidates $\{0.01, 0.05, 0.1, 0.2, 0.5\}$.

The second baseline algorithm is LSALR which uses $\eta \cdot (1 + \log(1 + 1/\|\delta \boldsymbol{W}_l\|_2))$ as the learning rate for the layer $l$. The global learning rate is set to be $\eta = 0.1$, which achieves best performance comparing from a pool of candidates $\{0.006, 0.05, 0.1, 0.2, 0.5\}$.

The third baseline algorithm is LARS which uses $\eta \cdot \frac{\|\boldsymbol{W}_l\|_2}{\|\delta \boldsymbol{W}_l\|_2}$ as the learning rate for layer $l$. In our experiment, we use the global learning rate $\eta = 2$ for LARS, which achieves best performance from a pool of $\{0.1, 1, 2, 5, 10\}$.

For baseline algorithms, we apply *weight decay* with coefficient *1e-3* if without specific description.

At last, we present the test accuracy of different VGG nets for classification of CIFAR-10 and CIFAR-100 in Table 1. We report the median of 3 independent runs of each pair of model and algorithm. For this group of experiments, we use global learning rate $\eta = 0.1$ and weight decay coefficient *5e-3* for our algorithm. Our algorithm achieves higher test accuracy over its competitors on all four VGG models with margins.

Table 1: Classification accuracies for CIFAR-10 and CIFAR-100.

| | CIFAR10 | | | | CIFAR100 | | | |
|---|---|---|---|---|---|---|---|---|
| | VGG11 | VGG13 | VGG16 | VGG19 | VGG11 | VGG13 | VGG16 | VGG19 |
| SGD | 92.34 | 93.90 | 93.72 | 93.47 | 71.84 | 74.07 | 72.86 | 71.35 |
| LARS | 91.81 | 93.40 | 93.47 | 93.48 | 67.26 | 70.35 | 69.90 | 69.52 |
| LSALR | 92.58 | 93.68 | 93.35 | 93.46 | 71.14 | 73.74 | 73.14 | 70.76 |
| OURS | 92.45 | **94.11** | **93.90** | **93.88** | **73.39** | **75.32** | **74.68** | **72.82** |