[Reviews · NeurIPS 2018]

Reviewer 1



This paper uses the concept or "back-matching" (target propagation) as a way to analyze deep networks, and particularly resnets. After showing that the standard gradient descent correspond to block-gradient step of back-matching, the authors use this to analyze the convergence of deep networks. They link the slow convergence of networks to ill-conditioned Hessian matrices of the blocks, using theoretical and empirical arguments, and linking to previously published papers. They explain how skip connections and batch normalization produce better conditioned Hessians, which could explain their success. Finally, they use this analysis to derive a simple variant of SGD that improves the conditioning of the Hessian and produces better empirical results. Pros: - This paper is well written, clear and cites previous work. - The paper uses sound theoretical arguments to explain things that have been observed in other works. - The author derive a simple algorithm that indeed produces good results, which is an indicator that the theory and insights are likely to be correct. Besides, this algorithm (scale-amended SGD) can be used in practice. Cons: - The experimental results could be stronger. Although many experimental results this paper cites come from ImageNet, their own experiments are only run on CIFAR-10 and CIFAR-100. Therefore, it is unclear if scale-amended SGD will be useful in practice on large dataset.

Reviewer 2



The authors introduce a training algorithm (back-matching propagation) which optimises each layer of a neural network using a separate matching loss. This is inspired by the work of Carreira-Perpinan and Wang and their Method of Auxiliary Coordinates. This approach allows the authors to study the Hessian of the loss at each layer, with respect to the parameters at that layer. These are the local Hessians. Efficiency of Deep Learning tricks such as skip connections and batch normalization can then be studied using these local Hessians. The authors also propose a variant of SGD, called scale amended SGD, and test this variant in an experiment. Some issues with the paper: * Comparison of vanilla SGD and scale amended SGD in section 4 is done using a single run of training with each algorithm. There is no confidence interval. * How where the hyper-parameters chosen for Figure 1 in section 4? Since this compares two different algorithms, there is no guarantee that a set of optimal parameters for one algorithm are also optimal for the other. How do we know that different hyper-parameters would not show better results for vanilla SGD? Other comments: * In section 2, neither Procedure 1 nor formula (5) define the update rule for $z^{k+1}$. This needs to be clarified. * If $k$ is an index for the various elements in the input data $X$, what is the meaning of the notation $z_b^{k+\frac{1}{2}$? * Footnote 3 on page 4 should be moved earlier in the paper since bias terms have already been removed in previous formulas. ===== Thank you to the authors for their reply. Their response do address the points I have raised. I am happy to raise my review to 'marginally above' provided the authors do the following: 1) make an effort to better explain what their algorithm exactly is (in particular, properly define all variables, with clear update rules); 2) replace their single run curve with the multiple run graph they put in their rebuttal.

Reviewer 3



#### Summary The authors attempt to characterize neural network architectures and training methods conducive to successful training. They propose that backpropagation with respect to a loss function is equivalent to a single step of a "back-matching propagation" procedure in which, after a forward evaluation, we alternately optimize the weights and input activations for each block to minimize a loss for the block's output. The authors propose that architectures and training procedures which improve the condition number of the Hessian of this back-matching loss are more efficient and support this by analytically studying the effects of orthonormal initialization, skip connections, and batch-norm. They offer further evidence for this characterization by designing a blockwise learning-rate scaling method based on an approximation of the backmatching loss and demonstrating an improved learning curve for VGG13 on CIFAR10 and CIFAR100. #### Quality The authors assembled a good set of data points to make the case for investigating backmatching propagation procedures, but the strength of this case may be insufficient for a complete work at NIPS. The authors acknowledged that one of the assumptions used for the residual connections case may be too strong and that empirical validation is lacking. Additionally, the scale-amended SGD experiment results appear to rely on individual training runs. For experiments at this scale, it seems reasonable to expect some measure of variance from multiple trials. #### Clarity - If I understood the paper's case correctly, the claim that improving the training efficiency requires or implies a well-conditioned local Hessian is not mathematically strict and rather depends on the collection of observations listed above. If this is the case, the authors should consider softening some assertions in the abstract. - I believe equation (12) somewhat overloads F_b; would it be correct to rewrite this as $z_b = F_b(W_b, z_{b-1}) = z_{b-1} = \phi_b(W_b, z_{b-1})$? - In Section 4, it's not clear how m_{b, W} and m_{b, z} are computed for Algorithm 2. #### Originality Variants of back-matching propagation are explored in prior works but these did not apply this procedure to study the efficiency of vanilla backpropagation. #### Significance A general characterization of efficiency for deep learning architectures and training procedures would be very valuable, and the proposal offered here appears to be worth further study.